# Magnetic Nanodiscs—A New Promising Tool for Microsurgery of Malignant Neoplasms

**DOI:** 10.3390/nano11061459

**Published:** 2021-05-31

**Authors:** Tatiana N. Zamay, Vladimir S. Prokopenko, Sergey S. Zamay, Kirill A. Lukyanenko, Olga S. Kolovskaya, Vitaly A. Orlov, Galina S. Zamay, Rinat G. Galeev, Andrey A. Narodov, Anna S. Kichkailo

**Affiliations:** 1Laboratory for Biomolecular and Medical Technologies, Krasnoyarsk State Medical University Named after Prof. V.F. Voino-Yasenecky, 660029 Krasnoyarsk, Russia; zamaytn@krasgmu.ru (T.N.Z.); k.a.lukyanenko@yandex.ru (K.A.L.); olga.kolovskaya@gmail.com (O.S.K.); galina.zamay@gmail.com (G.S.Z.); 2Laboratory for Digital Controlled Drugs and Theranostics, Federal Research Center, Krasnoyarsk Science Center Siberian Branch of Russian Academy of Science, 660036 Krasnoyarsk, Russia; 3Institute of Physics and Informatics, Astafiev Krasnoyarsk State Pedagogical University, 660049 Krasnoyarsk, Russia; plufe@yandex.ru; 4Molecular Electronics Department, Federal Research Center, Krasnoyarsk Science Center Siberian Branch of Russian Academy of Science, 660036 Krasnoyarsk, Russia; sergey-zamay@yandex.ru; 5School of Fundamental Biology and Biotechnology, Siberian Federal University, 79 Svobodny pr., 660041 Krasnoyarsk, Russia; 6School of Engineering Physics and Radio Electronics, Siberian Federal University, 79 Svobodny pr., 660041 Krasnoyarsk, Russia; orlhome@rambler.ru; 7Kirensky Institute of Physics Federal Research Center KSC Siberian Branch Russian Academy of Sciences, Akademgorodok 50, bld. 38, 660036 Krasnoyarsk, Russia; 8JSC «NPP «Radiosviaz», 660021 Krasnoyarsk, Russia; krtz@mail.ru; 9Traumatology Orthopedics and Neurosurgery Department, Krasnoyarsk State Medical University Named after Prof. V.F. Voino-Yasenecky, 660029 Krasnoyarsk, Russia; narodov_a@mail.ru

**Keywords:** nanodiscs, microdiscs, magnetomechanical therapy, magnetic field, the nanoscalpel

## Abstract

Magnetomechanical therapy is one of the most perspective directions in tumor microsurgery. According to the analysis of recent publications, it can be concluded that a nanoscalpel could become an instrument sufficient for cancer microsurgery. It should possess the following properties: (1) nano- or microsized; (2) affinity and specificity to the targets on tumor cells; (3) remote control. This nano- or microscalpel should include at least two components: (1) a physical nanostructure (particle, disc, plates) with the ability to transform the magnetic moment to mechanical torque; (2) a ligand—a molecule (antibody, aptamer, etc.) allowing the scalpel precisely target tumor cells. Literature analysis revealed that the most suitable nanoscalpel structures are anisotropic, magnetic micro- or nanodiscs with high-saturation magnetization and the absence of remanence, facilitating scalpel remote control via the magnetic field. Additionally, anisotropy enhances the transmigration of the discs to the tumor. To date, four types of magnetic microdiscs have been used for tumor destruction: synthetic antiferromagnetic P-SAF (perpendicular) and SAF (in-plane), vortex Py, and three-layer non-magnetic–ferromagnet–non-magnetic systems with flat quasi-dipole magnetic structures. In the current review, we discuss the biological effects of magnetic discs, the mechanisms of action, and the toxicity in alternating or rotating magnetic fields in vitro and in vivo. Based on the experimental data presented in the literature, we conclude that the targeted and remotely controlled magnetic field nanoscalpel is an effective and safe instrument for cancer therapy or theranostics.

## 1. Introduction

Malignant neoplasms remain one of the leading causes of mortality in the working-age population [1] and are an important public health problem [2]. Despite the development of new methods for diagnosing oncological diseases and anticancer therapy, the proportion of deaths among all cancer patients exceeds 40%.

Currently, surgical resection and radiation therapy for tumors remain the leading physical methods of removing or destroying malignant neoplasms. The main disadvantage of these methods is their high invasiveness. Both surgery and radiation therapy damage the healthy tissue surrounding the tumor. This approach is dangerous, especially in the treatment of brain tumors. Radical resection is impossible, since single tumor cells are invisible. The main challenge in radiotherapy is maximizing the dose to the tumor while minimizing the damage to the surrounding healthy tissue. Low doses of ionizing radiation could cause adverse effects, including inflammation, fibrosis, atrophy, vascular damage, hormone deficiency, and secondary malignancies [3].

Therefore, there is a need for a novel targeted approach capable of total removal of tumor tissues with minimal damage to the healthy surroundings. This instrument (nanoscalpel) should have three main properties: (1) it should be miniature (nano- or micro-sized); (2) it should have affinity and specificity only for tumor cells; (3) it should be able to be remotely controlled. The creation of such a tool is only possible through nanotechnology, using nanomaterials with unique electronic, optical, and magnetic properties. A nanoscale surgical instrument for tumor destruction should include at least two components (Figure 1). The first component (the micro- or nanoscalpel itself) should be capable of damaging the tumor cell under the influence of external forces, causing the processes of its death. The second component must act as a recognizing element and interact only with its target, thus bringing the nanoscalpel into contact with the tumor cell. Additionally, the nanoscalpel should have the possibility of being used with tumor cell imaging molecules, for example a fluorescent dye or an antitumor drug.

## 2. Nanoscalpel

Various physical forces can be used to destroy tumor cells, including thermal energy [4], mechanical energy [5], and ionizing radiation [5]. Physical forces trigger mechanisms of tumor cell death via necrosis or apoptosis. The mediator between external forces and the cell should be a structure that transforms one type of energy into another. Magnetic micro- and nanoconstructions are the most suitable structures that convert the energy of a magnetic field into mechanical energy. They can be manipulated in a variety of ways, through motion control, concentration, rotation, oscillation, assembly, or disassembly, using weak magnetic fields [6].

The magnetic field energy transformation is the most promising option for the destruction of tumor cells, since magnetic fields are safe for humans, and at the same time they can penetrate into the body and control magnetic structures [7]. Depending on the characteristics of the magnetic field, its energy can be converted into either heat energy, causing the particles heating and hyperthermia of the tumor tissue [4], or into mechanical energy, causing the oscillation or rotation of the magnetic particles, leading to the mechanically induced cell programmed death or necrotic damage [8,9].

In order to perform direct mechanical damage, the torque of the magnetization must be translated into a force applied to the cell, which is capable of damaging the cell membrane [10,11] or organelles (in case of penetration of magnetic particles into the cell) [7,12,13,14,15].

The magnetic field creates magnetization, which due to magnetic anisotropy, becomes a torque for a physical particle. This torque can be used in a variety of ways. The most attractive way to destroy tumor cells for microsurgery is magnetomechanical destruction due to a magnetic field’s action. Magnetomechanical transduction can act on death receptors [16], ion channels [17], or can directly damage the cells [10,13,15]. This effect has been shown both in vitro [10,14] and in vivo [7,12].

That is why recently in biomedical research, magnetic nanoparticles have begun to be actively used. These magnetic nanoparticles are very diverse: (1) they can have different sizes and structures; (2) can be homogeneous or consist of several layers; (3) can have different magnetic properties, depending on the chemical composition, the type of the crystal lattice, and their interactions with neighboring particles [6].

Superparamagnetic iron oxide nanoparticles (SPION), which exhibit magnetic properties only when a magnetic field is applied, are the most popular ones [15]. Without a magnetic field, the magnetic moment of the nanoparticles equals zero. Such nanoparticles are often used for magnetic resonance imaging, hyperthermia induction, and mechanical destruction of cells [4,15,18]. However, the effectiveness of these nanoparticles in destroying tumor cells has reached its limit [19]. Their size restrict the magnitude of the SPION magnetic response required for biomedical applications, since nanoparticles aggregate above the superparamagnetic threshold [19].

Usually, magnetic nanoparticles (MNPs) have a magnetic core, protective shell, and a biologically functional coating. Ferromagnets, ferrimagnets, and superparamagnets are used as magnetic materials. Various protective materials are used for MNP coatings for biomedical research, since uncoated magnetic particles are unstable under physiological conditions [20,21]. In addition, they contribute to the formation of free radicals [22], can agglomerate [23], and can be opsonized and captured by macrophages [24].

For targeted delivery, MNPs are functionalized with ligands capable of specifically binding to target cells. Peptides, antibodies, aptamers, and small molecules with affinity and selectivity are used as carriers. Ligands increase the circulation time of magnetic particles in the blood and improve their biocompatibility [25,26,27].

The use of a nanoparticle-based magnetomechanical approach using a low-frequency, non-heating magnetic field demonstrated the ability of magnetic nanoparticles to convert the energy of a magnetic field into deformation and to create a change in the conformation of macromolecules attached to them [28]. Thus, MNPs turned out to be capable of manipulating cellular functions with mechanotransduction using remote control of the magnetic field [29,30]. In magnetic fields, MNPs can stimulate or suppress cellular functions such as apoptosis, differentiation, migration, proliferation, and secretion [31,32]. An alternating magnetic field can cause oscillations of magnetic nanoparticles and can form the basis of magnetomechanical remote destruction of tumor cells [30]. Successful destruction of tumor cells using MNPs functionalized with antibodies under the action of an alternating magnetic field has been demonstrated in vitro and in vivo [33]. Other studies have demonstrated alternating magnetic fields driving the magnetomechanical in vivo tumor destruction with aptamer-functionalized MNPs [34]. However, superparamagnetic nanoparticles are primarily used to induce hyperthermia in tumor tissues [30,35,36,37,38]. This approach is based on the high sensitivity of tumor tissues to slight increases in temperature. Heating the tumor up to only +42 °C causes irreversible disruption of the protein conformation due to the tumor tissue’s higher acidity. In contrast, the proteins of normal tissues are insensitive to this temperature [39]. From a physical point of view, MNPs convert magnetic energy into heat during their magnetization reversal in high-frequency magnetic fields. Magnetic nanoparticles suitable for hyperthermia have a high specific absorption rate (SAR), allowing them to heat up quickly in alternating magnetic fields. However, the efficiency of converting magnetic field energy into thermal energy is low, and the MNP doses or magnetic anisotropy have to be increased [40,41].

An increase in the magnetic moment of the nanoparticle improves the efficiency of the nanoscalpel. However, an increase in the magnetic moment contributes to the undesirable effects of NP aggregation during the suspension of the preparation. In attempting to find a compromised state, it is essential to understand the physical mechanisms involved in forming the nanoparticles’ magnetic structure. Additionally, the chemical synthesis of superparamagnetic nanoparticles remains very difficult for mass production due to the relatively low yield and poor reproducibility of the quality of the nanoparticles.

The search for new concepts and strategies has shown that magnetic discs are the most promising magnetic structures for the magnetomechanical destruction of tumor cells. Magnetic discs represent a new generation of particles that can solve biomedical problems in cancer treatment. Magnetic micro- and nanodiscs are characterized by a high saturation of magnetization and the absence of residual magnetization, facilitating remote control of particles with a magnetic field. These properties help to avoid disc agglomeration and make the discs ideal magnetomechanical actuators for disrupting the integrity of cancer cells [20].

## 3. Properties of Magnetic Nano- and Microdiscs

To date, four types of magnetic discs are used for tumor cell destruction: synthetic antiferromagnetic SAF (in-plane) and P-SAF (perpendicular) [5,42,43,44], vortex Py [10], and three-layer non-magnetic–ferromagnetic–non-magnetic systems [45]. The pronounced magnetostrictive properties and small crystallographic anisotropy make nickel a preferable ferromagnetic layer in the disc’s center [17] (Figure 2). Magnetic discs used for tumor destruction have magnetic anisotropy. This is formed by the dependence of the magnetic properties of a ferromagnet on the direction of magnetization relative to the structural axes of the crystal. Weak relativistic interactions between atoms cause anisotropy, such as spin–orbit or spin–spin interactions.

The first type of synthetic antiferromagnetic (SAF) disc consists of two ferromagnetic layers separated by a non-magnetic layer. The magnetic layers become exchange-coupled and behave as single magnetic moments with opposite magnetization directions, compensating each other. When an external magnetic field is applied, the magnetic moments are oriented in one direction, reaching saturation. Synthetic antiferromagnetic nanoparticles have unique magnetic properties [43]. The thickness and material of the non-magnetic spacer control the exchange coupling between the ferromagnetic layers, leading to the magnetization of these layers in opposite directions. As a result, the structure with zero net magnetization easily switches in an external field. By varying the thickness of the layers that make up the SAF and obtaining magnetic susceptibility, it is possible to control the dispersion of particles in solution [46]. P-SAF discs are similar to SAF discs; however, their magnetization is perpendicular to the disc’s plane [47].

Vortex discs are characterized by the fact that magnetic moments are twisted in the disc plane and arranged in closed circles to minimize magnetostatic energy. Only in the core do the magnetic moments go out of the plane, where they are directed perpendicularly. This rotation pattern also has zero residual force. When an external magnetic field is applied, the vortex core is displaced in the disc plane until it reaches the edge where it is annihilated, while the disc becomes magnetically saturated [10].

To cause vibration or oscillation of the particles, the applied magnetic field should rotate or change with sufficient amplitude, preferably close to the particle saturation field. The key parameter is the particle’s magnetic anisotropy, which relates the torque of the magnetization to the mechanical torque on the particle. The magnetic discs’ diameter and thickness can be optimized to obtain the desired magnetic vortex configuration [48]. A planar magnetic structure would provide low residual magnetization, sufficient magnetic susceptibility for the effective magnetic actuation, and insufficient agglomeration [44,49,50].

Nano- and microdiscs are made using methods that combine lithography (usually photolithography or electron beam) and physical deposition of materials (spraying or thermal evaporation). The ease of controlling disc manufacturing conditions makes the process automatic and repeatable. The shape and ease of manufacture, including single-stage metal deposition on a patterned photoresist layer, allows the discs to be reduced to 100 nm in size while maintaining the vortex state [51,52]. Disc sizes can be further reduced by using the latest advances in nanoimprinting and deep UV lithography [53]. The methods used for the manufacture of synthetic antiferromagnetic and vortex discs and discs with a flat quasi-dipole magnetic structure make it possible to strictly control their shape, size, and composition, making the manufacturing process reproducible.

## 4. Biological Effects of Discs on Tumor Cells in an Alternating or Rotating Magnetic Field

A fundamental property of all living organisms is mechanosensitivity, which underlies exo- and end reception. This reception controls the parameters of the internal and external environments of the organism. Mechanoreceptors control the cells’ functional state, tissue growth, differentiation of stem cells, apoptosis, and necrosis [54,55,56]. Therefore, the mechanosensitivity of cells, including tumor cells, provides an opportunity for external control of their functional state.

Vortex and artificial antiferromagnetic (SAF, P-SAF) magnetic nano- and microdiscs and discs with a flat quasi-dipole magnetic structure are promising tools for the remote control of the functional state of cells. In a low-frequency rotating or alternating magnetic field, discs can convert magnetic field energy into mechanical energy. At present, micro-and nanodiscs with zero magnetization have begun to be used in experiments to modulate cell function in the course of bone tissue regeneration [32] and destruction of tumor cells [10].

Data on the ability of vortex and artificial antiferromagnetic discs and discs with a flat quasi-dipole magnetic structure to stimulate the death of tumor cells under the influence of a magnetic field in vitro and in vivo are summarized in Table 1. Discs have zero total magnetization in the absence of a magnetic field and high magnetization under weak external magnetic fields [10]. This property ensures the high sensitivity of discs to magnetic stimuli. Various magnetic field parameters are used for magnetomechanical therapy; in particular, the magnetic field strength varies from 5 mT to 1 T, and the frequency of the alternating field is 10–50 Hz. Therapy duration ranges from 1 min to 2 h. At the same time, the biological effect of the applied therapy is practically the same (Table 1). Apparently, a short time is sufficient for the destruction of tumor cells using nanodiscs in an alternating magnetic field.

Kim et al. showed for the first time that in a rotating magnetic field, vortex discs (20:80 iron-nickel (permalloy) coated with 5 nm Au) destroy 90% of human glioma tumor cell line No. 10 in vitro [10]. Subsequently, the ability of vortex microdiscs to promote tumor cell death was demonstrated by other authors in vitro and in vivo [7,14,17]. Simultaneously, vortex nanodiscs (140 nm) were more effective in the destruction of tumor cells than vortex microdiscs (1 μm) [57].

The ability to stimulate tumor cell death under the influence of magnetic fields was found in the vortex and antiferromagnetic discs (SAF and P-SAF). Comparing the efficiency of a vortex and P-SAF discs to destroy tumor cells, Mansell et al. showed [53] that P-SAF discs are five times more efficient than vortex discs. The calculations showed that in the case of a rotating magnetic field, microdiscs with uniaxial anisotropy provide continuous torque, in contrast to microdiscs with a plane of easy magnetization, which happens when this effect occurs temporarily when the field is applied.

The study of the dependence of the magnetic discs’ efficiency on the frequency of the applied magnetic field showed that the optimal effect is achieved at frequencies of 10 and 20 Hz (∼90% of cell death). Increasing the frequency to 40 Hz reduced the cell mortality rate to ~75%, while at 50 Hz the cell mortality rate was only ~25%; at 60 Hz, the effect of cell death was absent [10]. The effect of lower frequencies (from 1 to 20 Hz) was evaluated by Wong et al. [49], who showed a slight increase in the lethal effect, with a decrease in frequency from ~80% of viable cells at 10 Hz to ~73% of viable cells at 1 Hz.

## 5. The Mechanism of Tumor Cell Death Exposed to Discs under the Influence of a Magnetic Field

The death of tumor cells using micro- and nanodiscs can occur as a result of necrosis or apoptosis. Necrosis occurs due to mechanical destruction of the cell membrane (magnetoporation) or destruction of the entire cell (magnetolysis). Necrosis is an unfavorable method of tumor destruction, since it causes inflammation due to lysosomal enzyme release. Therefore, the most favorable tumor cell elimination is apoptosis, which is suppressed in cancer cells due to oncogenic mutations. The inflammatory process does not accompany apoptosis; therefore, all researchers are looking for magnetic field characteristics stimulating apoptosis in tumor cells [10,17]. In the magnetic field, discs induce cell death in two ways: (1) by internalizing into the cell (Figure 3a); (2) by acting on the cell through the proteins of the cell membrane (Figure 3b).

## 6. Internalization of Magnetic Discs

The penetration of magnetic discs into the cells is carried out by endocytosis with the subsequent encapsulation into lysosomes. It is assumed that the rupture of the lysosomal membrane under the influence of a magnetic field, in this case, is the main cause of cell death [43]. At the same time, the internalization of the discs in the absence of a magnetic field does not reduce tumor cell viability. The internalization of the discs begins with their effects on the cell membrane, possibly by binding to membrane proteins containing many functional groups that can react with the discs’ surfaces. Discs internalized in the cell accumulate in lysosomes containing hydrolytic enzymes. The number of discs penetrating the cell depends on their size. Studies have shown that on average, each cell accumulates around ten microdiscs or one hundred nanodiscs. The calculation showed that the mass of permalloy (Ni80Fe20) absorbed by cell composition nanodiscs is almost two orders of magnitude lower than the mass of permalloy absorbed by the cell composition microdiscs. Moreover, the number of magnetomechanical drives—disc-shaped particles with a magnetic vortex state—is ten times higher. Thus, with a greater biological effect, nanodiscs exhibit less toxicity.

## 7. Impact of Discs on the Cell Membrane

The effects of the discs on the cell membrane of tumor cells and their internalization into the cell occur due to their binding to membrane proteins. This effect of the magnetic discs on tumor cells induces programmed cell death (apoptosis) [10,58,59]. The binding of discs to the cell membrane occurs due to their functionalization via recognition molecules—either antibodies or aptamers. Antibodies to carbonic anhydrase CA9, which is overexpressed on the cell surfaces of solid tumors, were used to functionalize the discs. Magnetic discs with anti-CA9 induced apoptosis in 70% of tumor cells under the influence of a magnetic field [17]. Using magnetic discs functionalized with antibodies to IL13α2R, other authors changed the homeostasis of calcium cations in tumor cells, stimulating apoptosis [10]. In this case, the stimulation of mechanosensitive cell receptors using vibrating discs caused the cell membrane to stretch, which led to an increase in the intracellular level of calcium cations [59]. Presumably, the cytoskeleton, which responds to external or internal physical stimuli and is responsible for cellular mechanical signal transmission, cell shape regulation, and migration, plays a key role in controlling the cell’s functional state in external magnetic fields. A schematic diagram of the target cell’s functional state is shown in Figure 4, showing the action of magnetic discs on membrane proteins remotely controlled by an alternating or rotating magnetic field.

## 8. Biological Effect of Magnetic Discs In Vivo

The biological effects of magnetic discs under the influence of the magnetic field, observed in vitro, are often different from the in vivo effects. When the discs enter the blood, they appear in a complex multi-component medium, the parameters of which are different from the cell culture conditions. For this reason, to achieve the required biological effects with magnetic discs, it is necessary to consider the behavior of the discs at the cell and organism levels, including their behavior in the bloodstream and tumor tissue. An analysis of the literature by Wilhelm et al. [60] showed that only 1% of nanoparticles introduced into the organism enter solid tumor cells. At the same time, most of them accumulate in the liver, spleen, and lungs [60]. Small nanoparticles are excreted by the kidneys, lymph nodes, and skin. However, the aptamers’ functionalization provides nanoparticle-targeted delivery to the tumor site and increases biocompatibility [27].

## 9. Transfer of Magnetic Discs along with the Bloodstream

In the bloodstream, anisotropic magnetic discs tend to deflect towards the vessel wall and carry out lateral drift along streamlines to the endothelium due to inertial and hydrodynamic forces [61]. This leads to the adhesion of particles to the endothelium near the tumor site, facilitating their diffusion from the blood vessels into the tumor tissue according to the “effect of increased permeability and retention (EPR)” in the tumor.

The tumor tissue has an altered vasculature [60] (Figure 5). Tumor mother vessels are characterized by thinning or contraction of endothelial cells, basal membrane degradation, and pericytes detachment (Figure 5c1). As a result of these structural changes, tumor vessels become highly permeable for small and large molecules and particles. These vessels are unstable and eventually differentiate into glomeruloid microvascular proliferation (Figure 5c2), vascular malformations (Figure 5c3), and capillaries (Figure 5c4). Glomeruloid microvascular proliferations are tortuous vessels composed of irregular layers of pericytes and endothelial cells with multiple very small vascular lumens (Figure 5c2). Vascular malformations are similar in size to the mother vessel but with a smooth muscle covering (Figure 5c3). Capillaries are formed from mother vessels and glomeruloid microvascular proliferation (Figure 5c4). Mother vessels can also include holes and open pores through which macromolecules can extravasate. In the extravasation mechanism of interendothelial cells, nanoparticles are transported through gaps measuring 100–500 nm in diameter [60].

The anisotropic shape of the magnetic particle facilitates its better penetration into the tumor [62]. The vibration of anisotropic particles due to hydrodynamic or magnetic forces causes enhanced particle interactions with the vessel wall and transmigration into the tumor [43,62]. The vascular density of a tumor is usually the highest at the tumor–host interface. Central parts of tumors tend to be less vascularized, which often have zones of necrosis due to insufficient blood supply. Plasma proteins increase the viscosity of the blood, slowing down the blood flow. For this reason, nanoparticles within the tumor vessels tend to move slowly or stagnate. Therefore, nanoparticles have enough time to diffuse from the vessel into the tumor’s extracellular matrix. Size is an important parameter for magnetic particles—they should be small enough not to clog blood microcapillaries and pass through the pores of blood vessels in order to diffuse into tissues.

## 10. Modification and Functionalization of Discs

The stability of magnetic particles in suspension depends on hydrophobic–hydrophilic and van der Waals forces. Magnetic discs have a large surface area to volume ratio, capable of forming micrometer-sized clusters [43]. To reduce cluster formation, surface modification with surfactants, natural dispersants, organic dyes, or polymers is required. This makes magnetic particles more stable in biological media. The dispersibility of magnetic particles can be improved using silica and gold. However, the use of non-magnetic materials to coat magnetic particles can lead to a decrease in magnetization saturation. To optimize the characteristics of disc-shaped magnetic particles, a compromise should be found between surface modification and retention of magnetic properties [43].

The stability of magnetic discs could also be increased through their functionalization with specific ligands, promoting targeted action on a tumor cell. Additionally, to reduce side effects, the magnetic particles must only enter the tumor cells. Additional functionalization with the ligands specific only to tumor cells, such as vascular endothelial growth factor receptor (VEGFR), transferrin receptors, or integrins, is required for the targeted action [63,64]. Additionally, the optimization of the target biodistribution of magnetic discs in vivo will be determined by the local blood flow, pH, organization of the vascular network, and extracellular matrix.

Consequently, specific targeting is necessary to increase the efficiency of the delivery of magnetic discs to the tumor. This can be achieved by using ligands that are complementary to target sites, which can be peritumoral and intratumoral blood vessels, the extracellular matrix, tumor cells, or intracellular targets with non-specific (passive) aim. To perform functionalization, the surfaces of nanoparticles are covered only with stabilizing agents. The functionalization of magnetic particles with ligands specific for tumor cells will increase their accumulation in the tumor tissues. The biological effects of magnetic discs functionalized with tumor cell-specific ligands (antibodies or aptamers) are shown in Table 2.

Aptamers are one of the preferable molecules as ligands for targeting due to the following properties: (1) aptamers can be selective to any desired target; (2) similarly to antibodies, aptamers have high specificity and high affinity to their target; (3) aptamers are manufactured chemically in an easily scalable process; (4) the chemical process for the production of aptamers is not susceptible to viral or bacterial contamination; (5) aptamers are non-immunogenic and non-toxic; (6) the small size of the aptamers allows them to penetrate effectively into any tumors; (7) aptamers can reversibly denature with the restoration of the desired conformation, while the phosphodiester bond is chemically stable; (8) aptamers can be chemically modified without changing their conformation; (9) chemical compounds for the addition of dyes or functional groups can be easily introduced during synthesis [65].

## 11. Toxicity of Magnetic Discs

The low toxicity and biocompatibility of magnetic discs is a prerequisite for creating a nanoscalpel for tumor microsurgery. Their chemical composition primarily determines the toxicity of MNPs. Iron-based MNPs are less toxic, since iron is easily degraded in the body. Manganese and zinc are more harmful than iron and are practically not used without preliminary surface modification [66]. Cobalt and nickel used in magnetic hyperthermia are highly toxic and require a special coating [67]. The toxicity of MNPs depends on the chemical nature of their coating, their biodegradability, and the compatibility of the MNP surfactants with the environment [27].

The mechanism of the toxic effects of MNPs has not been entirely revealed; however, it is assumed that it is primarily caused by oxidative stress [68], caused by the formation of reactive oxygen species (ROS) in the Fenton reaction of Fe^2+^ + H_2_O_2_ = Fe^3+^ + OH• + OH^−^, when magnetite interacts with the cell [69,70]. Iron ions with ROS are formed during magnetite destruction in lysosomes. In addition, ROS can be generated from the MNP surfaces by leaching metal ions or releasing oxidants via enzymatic degradation of the MNP. The resulting hydroxyl radicals react with DNA, proteins, polysaccharides, and lipids. ROS accumulation destroys cellular proteins, enzymes, lipids, and nucleic acids, contributes to cellular metabolism disruption and leading to apoptosis and necrosis [71]. Low concentrations of superparamagnetic iron oxide nanoparticles prevent cells from the damages and death caused by oxidative stress [72,73]. In particular, 10 nm nanoparticles size at concentrations of 10–30 μg/mL did not cause cytotoxicity in HeLa cells and did not affect the rats’ behavior, and were not toxic to the liver, kidneys, lungs, or spleen [74]. Fe_3_O_4_ @ Au MNPs measuring 18 nm in diameter and stabilized by citrate ions at concentrations ranging from 50 μg/mL to 2 mg/mL, did not reduce the survival of human liver carcinoma cells [75]. MNPs measuring 5–6 nm in size and stabilized with citrate ions, PEG, and glucose at concentrations of up to 1 mg/mL did not affect the survival of HeLa cells [76], while 35 nm Fe_3_O_4_ @ Au MNPs stabilized by citrate ions at 1 mg/mL concentration did not affect the survival of mouse fibroblasts [77]. It was found that MNPs enter cells without disrupting the membrane or the cytoskeleton. In general, the toxic effects of nanosized particles are caused by their high reactivity, their effective diffusion through biological membranes, and the ability to overcome tissue barriers [69].

Magnetic fields are attractive for therapy, since they can penetrate the entire depth into the internal organs and tissues of the body. The biosafety of magnetic fields for magnetodynamic therapy is essential. Most fields used for theranostics are practically safe [78].

Numerous studies have shown that magnetic nano- and microdiscs internalized into a cell, regardless of their number in the absence of a magnetic field, do not exhibit cytotoxicity and do not affect cell viability [5,20]. However, the data on the effects of magnetic discs on cell proliferation are ambiguous.

The selective action of nanodiscs and MNPs on target cells due to aptamers significantly increases their biocompatibility and reduces their toxicity [27].

Thus, it can be concluded that microsurgery on malignant tumors using a nanoscalpel based on functionalized tumor-recognizing ligands is low in toxicity.

## 12. Conclusions

The success of oncological disease therapies is determined by the efficiency of removing all transformed cells characterized by uncontrolled growth and division from the body. Surgical removal of tumors is one of the most popular methods of treating malignant neoplasms. In this case, the organism does not eliminate the dead tissue, such as during radiation or chemotherapy. However, there is a high probability that not all tumor cells will be removed during the surgery. In contrast, healthy cells may be damaged, which is especially dangerous during surgical treatment of glial brain tumors. The majority of tumors metastasize, causing secondary tumors. These factors demonstrate the need to develop a surgical instrument with new properties (a nanoscalpel), which works according to the “find and neutralize” principle. The nanoscalpel should be in the nanoscale (for the destruction of individual tumor cells) (i), remotely controlled using forces that are safe for the body (ii), and targeted (to destroy only tumor cells without damaging healthy ones) (iii).

The proposed nanoscalpel, the structure of which consists of magnetic nanodiscs with anisotropy and the ability to convert the magnetic moment into mechanical torque under the influence of a safe low-frequency alternating or rotating magnetic field of low intensity, along with the targeting of tumor cells (due to a recognition ligand, either an antibody or aptamer), may become the most promising surgical instrument, and has already shown its effectiveness. One of the essential advantages of nanodiscs in comparison with spherical nanoparticles is their anisotropy, which facilitates their movement along the periphery of the vessel, due to which the probability of their passage through the endothelium into the tumor increases.

According to the literature, the most preferred nanodiscs are P-SAF antiferromagnetic discs, while in three-layer non-magnetic–ferromagnet–non-magnetic systems, nickel is used as a ferromagnetic filling due to its pronounced magnetostrictive properties in combination with the small constant of the crystallographic anisotropy. Aptamers are considered the most preferred targeting molecules because they are small in size. The nanoscale of the aptamers allows the nanoscalpel functionalized with this ligand to penetrate the tumor more effectively. Such characteristics are not available for large antibodies. Remote control by alternating or rotating magnetic fields could cause direct destruction of tumor cells. This treatment becomes especially valuable when the accumulation of discs in the tumor is visualized and confirmed using MRI.

Thus, the nanoscalpel, which is remotely controlled by a magnetic field and visualized with MRI and is capable of targeted delivery, is a promising safe tool for the treatment and visualization of malignant neoplasms. Soon, this approach may solve the problem of the targeted treatment of cancer patients under visual control.

## Figures and Tables

**Figure 1 nanomaterials-11-01459-f001:**
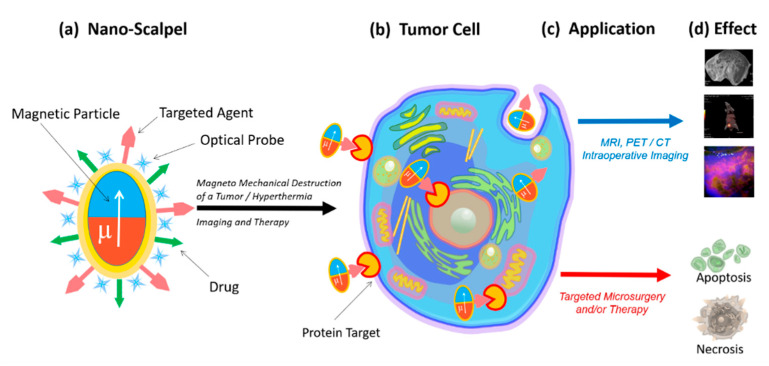
Schematic representation of the novel nanoscalpel device (**a**): binding selectively to tumor cells (**b**) for different applications (**c**) such as accurate diagnostics and targeted therapy (**d**).

**Figure 2 nanomaterials-11-01459-f002:**
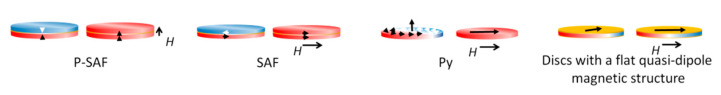
Magnetic discs are used to destroy tumor cells: synthetic antiferromagnetic P-SAF and SAF, vortex Py, and three-layer non-magnetic–ferromagnetic–non-magnetic systems with a flat quasi-dipole magnetic structure. P-SAF: Out-of-plane magnetic moments. Zero remanences. Under a magnetic field, the disc is magnetized with out-of-plane net magnetization. SAF: In-plane magnetic moments. Zero remanences. Under a magnetic field, the disc is magnetized with in-plane net magnetization. Vortex Py: In-plane magnetic moments. Zero remanences. Under a magnetic field, the disc is magnetized with in-plane net magnetization. Discs with a flat quasi-dipole magnetic structure: In-plane magnetic moments. Zero remanences. Under a magnetic field, the disc is magnetized with in-plane net magnetization.

**Figure 3 nanomaterials-11-01459-f003:**
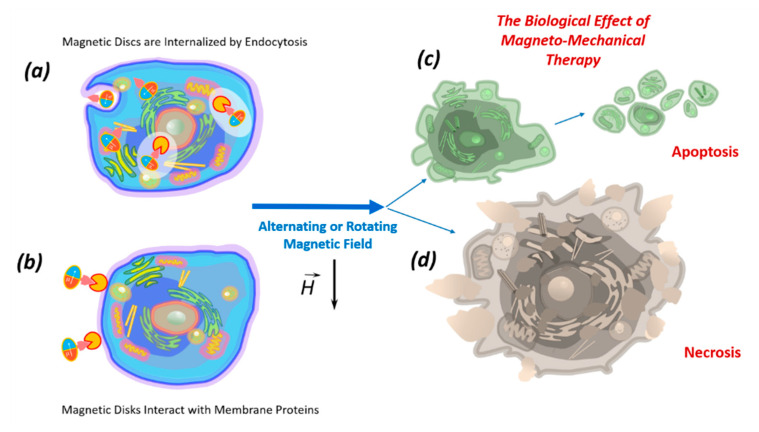
The biological effects of magnetic nano- or microdiscs functionalized by recognizing molecules, which in the low-frequency magnetic field act inside the cell after internalization by endocytosis (**a**) or directly influence the cellular membrane by interacting with membrane proteins (**b**), causing apoptosis (**c**) or necrosis (**d**).

**Figure 4 nanomaterials-11-01459-f004:**
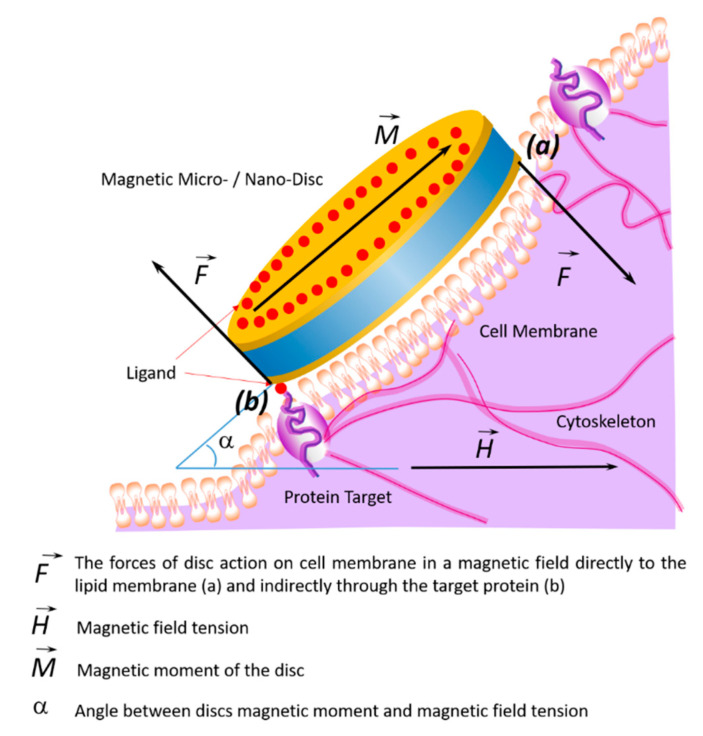
Schematic diagram showing changes of the target cell’s functional state under the influence of magnetic discs and the effects on membrane proteins in an alternating or rotating magnetic field. In a magnetic field, the discs’ forces act on the cell membrane and cytoskeleton in two ways: directly (**a**) or indirectly through the target protein (**b**).

**Figure 5 nanomaterials-11-01459-f005:**
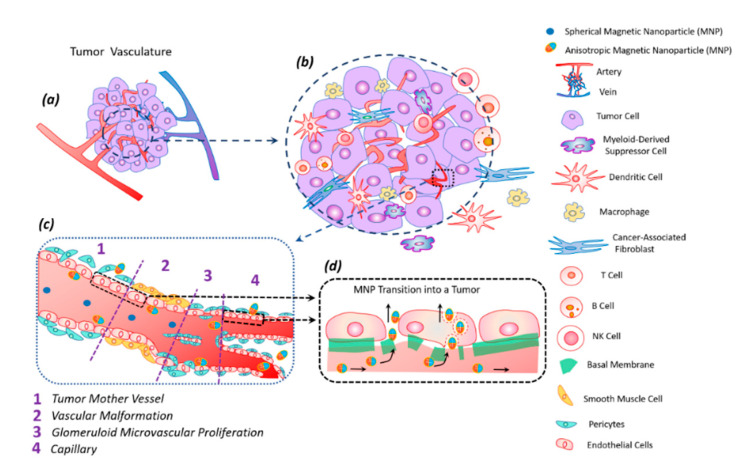
Tumor vasculature (**a**) in a complex tumor microenvironment (transformed and immune cells) (**b**). Transport of magnetic anisotropic discs and spherical magnetic nanoparticles along transformed tumor vessels (**c**). Spherical particles move in the central part of the vascular bed, while anisotropic magnetic discs move along the periphery of the vessel, frequently interacting with the vessel wall; this peculiarity facilitates discs transmigration into the tumor (**c**). Magnetic nanodiscs are able to pass into a tumor through the damaged basal membrane in two ways: through endothelial cells and the gaps between them (**d**).

**Table 1 nanomaterials-11-01459-t001:** The biological effect of magnetic antiferromagnetic (SAF and P-SAF) and vortex Py nano- and microdiscs and discs with a flat quasi-dipole magnetic structure.

Size	Disc Type	Composition	Magnetic Field Characteristics	Biological Effect In Vitro	Reference
2 μm	P-SAF	CoFeB connected by Pt/Ru/Pt spacers	Rotating magnetic field, 10 kOe Duration 1 min Torque 18 nN	Destruction of 62% of U87 cells	[53]
2 μm	Py	Ni_80_Fe_20_	Rotating magnetic field, 10 kOe Duration 1 min Torque 75 nN	Destruction of 12% U87 cells	[53]
150/200/350 nm	Py	Ni_80_Fe_20_	Alternating magnetic field, 20 HzDuration 2 h	Destruction of 83.4/83.2/82.5% of HeLa cells	[49]
2 μm	P-SAF	(Ta/Pt/CoFeB/Pt/Ru/ PT/CoFeB)_10_	Rotating magnetic field, 1 T Duration 20 min	Destruction of 70% of U87 cells	[7]
1.3 μm	Py	Ni_80_Fe_20_	Rotating magnetic field, ∼20–30 mT, 20 Hz Duration 1 h	Destruction of 70% of renal cancer cells	[17]
1 μm thickness 60 nm	Py	Ni_80_Fe_20_	Rotating magnetic field, 9 mT 20 Hz Duration 10 min	Destruction of 90% of human glioma tumor cell line No. 10 cells (*N*10)	[10]
2 μm	Py	Ni_80_Fe_20_	Rotating magnetic field 1 T, 20 Hz, Duration 30 min	Destruction of 60% of U87 cells. In vivo survival is 3 times higher, and the tumor is 3 times smaller	[7]
0.14 μm	Py	Ni_80_Fe_20_	Rotating magnetic field 10 mT, 20 Hz Duration 30 min	Destruction of 60% of cells	[57]
2 μm	Py	Ni_80_Fe_20_	Rotating magnetic field 10 mT, 20 Hz Duration 30 min	Destruction of 12% of cells	[57]
1 μm	Discs with a flat quasi-dipole magnetic structure	Au/Ni/Au	Rotating magnetic field, 50 Hz 5 mT Duration 20 min In vitro In vivo	Destruction of 80% of Ehrlich ascites adenocarcinoma cell	[18]
1 μm	Discs with a flat quasi-dipole magnetic structure	Au/Ni/Au	Alternating magnetic field, 50 Hz, 5 mT Duration 20 min	Destruction of 90% of Ehrlich ascites adenocarcinoma cell	[45]

**Table 2 nanomaterials-11-01459-t002:** The biological effects of magnetic discs functionalized with antibodies and aptamers.

Disc Type	Recognizing Agent	Disc Binding to Recognizing Agent	Cell Type; Destruction Rate	Reference
The 60-nm-thick, ~1-μm-diameter 20:80% iron–nickel (permalloy) discs, coated with a 5-nm-thick layer of gold on each side	Antibodies anti-IL13*α*2R	S–Au bond	Human glioma *N*10 cell line; 90% (in vitro)	[10]
The 60-nm-thick, ~1-μm-diameter 20:80% iron–nickel (permalloy) discs	Antibody antihCA9	S–Au bond	Renal SCRC-59 renal cancer line; 90% (in vitro)	[17]
Discs with a flat quasi-dipole magnetic structure Au/Ni/Au	Aptamer	S–Au bond	Ehrlich ascites adenocarcinoma cell line; 80% (in vitro, in vivo)	[18]
Discs with a flat quasi-dipole magnetic structure Au/Ni/Au	Aptamer	S–Au bond	Ehrlich ascites adenocarcinoma cell line; 90% (in vitro, in vivo)	[45]

## Data Availability

Data sharing not applicable. No new data were created or analyzed in this study. Data sharing is not applicable to this article.

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
