# Peer review of "Magnetic Nanodiscs—A New Promising Tool for Microsurgery of Malignant Neoplasms"

_nanomaterials, 2021, doi:10.3390/nano11061459_

Round 1

Reviewer 1 Report

The authors present a review of work relating to magnetic nano-scalpels. The content of the work is generally good and thorough. However, the use of English is poor to the point of being a distraction. It requires significant editing. Additionally, I have several minor critiques.

1) P-SAF and SAF are not defined at their first usage.

2) It would be extremely helpful to include a section that describes the different types of magnetic properties that nanoparticles can display and point out the relevant properties. 

3) "Under an H" is strange wording. I recommend changing it to under a magnetic field.  

4) The use of "mortality rate" is confusing in regard to cell culture (as in table 1). Table 1 generally needs to be edited for consistency and clarity. 

5) In table 2 it is unclear whether the mortality rate refers to in vitro or in vivo.

6) The toxicity section needs to be expanded. There is significant discussion on the topic in the literature and the nuance needs to be fleshed out. Additionally, it would be helpful to expand on off-target effects as well as any potential effects that result from the magnetic field in the absence of particles. 

Author Response

We thank the Reviewer for the valuable suggestions the manuscript has been corrected in accordance with comments.

The authors present a review of work relating to magnetic nano-scalpels. The content of the work is generally good and thorough. However, the use of English is poor to the point of being a distraction. It requires significant editing. Additionally, I have several minor critiques.

1) P-SAF and SAF are not defined at their first usage.

Thank you, this has been clarified.

2) It would be extremely helpful to include a section that describes the different types of magnetic properties that nanoparticles can display and point out the relevant properties. 

The properties of magnetic nanoparticles are presented in the section "Properties of magnetic nano- and microdiscs". The Section corrected.

3) "Under an H" is strange wording. I recommend changing it to under a magnetic field. 

Thank you for the comment. This has been corrected

4) The use of "mortality rate" is confusing in regard to cell culture (as in table 1). Table 1 generally needs to be edited for consistency and clarity. 

The table has been modified

5) In table 2 it is unclear whether the mortality rate refers to in vitro or in vivo.

The table has been modified

6) The toxicity section needs to be expanded. There is significant discussion on the topic in the literature and the nuance needs to be fleshed out. Additionally, it would be helpful to expand on off-target effects as well as any potential effects that result from the magnetic field in the absence of particles. 

The Section expanded. Added data on toxicity factors, mechanisms of toxicity and toxicity of the magnetic field.

Reviewer 2 Report

The subject is very interesting. Moreover, the review on this topic is very current. The analysis of the literature is meticulously performed.

However, I have some observations and recommendations to make to the authors. 

  1. I consider that the authors should carefully revise the text from the point of view of the clarity of expression, avoiding the repetition of some explanations and paying attention to the notations. For example, referring to paying attention to the notations, on page 3 the Authors' comments on superparamagnetic iron oxide nanoparticles (SPION). Further, the discussion refers to MNPs without any transition or explanation of the notation, MNPs. Also, this new paragraph: Supermagnetic nanoparticles for magneto-dynamic therapy, what it is? A new subsection of the subchapter Nanoscalpel?
  2. Another example, in the chapter Conclusions: The first phase is forgotten from the paper template. 

Author Response

We are grateful to the Reviewer for the comments and suggestions. 

1. I consider that the authors should carefully revise the text from the point of view of the clarity of expression, avoiding the repetition of some explanations and paying attention to the notations. For example, referring to paying attention to the notations, on page 3 the Authors' comments on superparamagnetic iron oxide nanoparticles (SPION). Further, the discussion refers to MNPs without any transition or explanation of the notation, MNPs. Also, this new paragraph: Supermagnetic nanoparticles for magneto-dynamic therapy, what it is? A new subsection of the subchapter Nanoscalpel?

The Changes have been made. The subsection name has been removed.

2. Another example, in the chapter Conclusions: The first phase is forgotten from the paper template. 

The text has been corrected.

Reviewer 3 Report

A systematic review of the  magneto-mechanical therapy was conducted from 2008 to 2020. The principle of this therapy is to apply a mechanical force on cancer cells, in order to destroy them, thanks to magnetic particles vibrations that are mechanically actuated by a non-thermal applied external magnetic field.

Magnetic particles of different physical constructions, remotely controlled by a safe “non thermal” magnetic stimulus, could act as a nanoscalpel  in the  tumor microsurgery. To destroy only tumor cells without damaging healthy ones, the surgical nanodevices could be functionalized with targeting molecules such as aptamers to target tumors.

The analysis of such studies shows the growing interest in this magneto-mechanical approach. The review reveals how the various type of magnetic micro–nanoconstructs used in the different studies – their shapes, sizes, compositions, the resulting magnetic states and properties – the non-agglomeration requirements and the advantages of magnetic anisotropy for an efficient mechanical actuation, have been detailed.

Authors could better detailed about the various available sources (frequency and amplitude)  of the applied rotating or alternating magnetic field. Furthermore, authors could detailed about the influence of magnetic field application time on cell viability.

Author Response

We thank the Reviver for the valuable comments and suggestions which helped to improve the text. The parameters of the magnetic field used by researchers for magneto-mechanical therapy differ; in particular, the magnetic field strength varies from 5 mT to 1 T, and the frequency of the alternating field is 10-50 Hz. The duration of therapy for different researchers ranges from 1 minute to 2 hours. At the same time, the biological effect of the applied treatment is practically the same (Table 1). It takes a short time and a weak magnetic field to trigger the destruction of tumor cells by magnetic disks under the influence of a magnetic field.
The frequency, intensity of the magnetic field, and duration of its exposure are presented in table 1.

Corrections have been made to the text. Figure 2 has been corrected.

Round 2

Reviewer 2 Report

Thank you for the very careful corrections you have made.